# A meta-analysis of threats to valid clinical inference in preclinical research of sunitinib

**Valerie C Henderson[1], Nadine Demko[1], Amanda Hakala[1], Nathalie MacKinnon[1], Carole A Federico[1], Dean Fergusson[2], Jonathan Kimmelman[1]\***

[1]Studies of Translation, Ethics and Medicine Research Group, Biomedical Ethics Unit, McGill University, Montréal, Canada; [2]Department of Clinical Epidemiology, Ottawa Hospital Research Institute, Ottawa, Canada

**Abstract** Poor study methodology leads to biased measurement of treatment effects in preclinical research. We used available sunitinib preclinical studies to evaluate relationships between study design and experimental tumor volume effect sizes. We identified published animal efficacy experiments where sunitinib monotherapy was tested for effects on tumor volume. Effect sizes were extracted alongside experimental design elements addressing threats to valid clinical inference. Reported use of practices to address internal validity threats was limited, with no experiments using blinded outcome assessment. Most malignancies were tested in one model only, raising concerns about external validity. We calculate a 45% overestimate of effect size across all malignancies due to potential publication bias. Pooled effect sizes for specific malignancies did not show apparent relationships with effect sizes in clinical trials, and we were unable to detect dose–response relationships. Design and reporting standards represent an opportunity for improving clinical inference.

**\*For correspondence:** jonathan. kimmelman@mcgill.ca

## Introduction

Preclinical experiments provide evidence of clinical promise, inform trial design, and establish the ethical basis for exposing patients to a new substance. However, preclinical research is plagued by poor design and reporting practices (*van der Worp et al., 2010*; *Begley, 2013a*; *Begley and Ioannidis, 2015*). Recent reports also suggest that many effects in preclinical studies fail replication (*Begley and Ellis, 2012*). Drug development efforts grounded on non-reproducible findings expose patients to harmful and inactive agents; they also absorb scarce scientific and human resources, the costs of which are reflected as higher drug prices.

Several studies have evaluated the predictive value of animal models in cancer drug development (*Johnson et al., 2001*; *Voskoglou-Nomikos et al., 2003*; *Corpet and Pierre, 2005*). However, few have systematically examined experimental design—as opposed to use of specific models—and its impact on effect sizes across different malignancies (*Amarasingh et al., 2009*; *Hirst et al., 2013*). A recent systematic review of guidelines for limiting bias in preclinical research design was unable to identify any guidelines in oncology (*Henderson et al., 2013*). Validity threats in preclinical oncology may be particularly important to address in light of the fact that cancer drug development has one of the highest rates of attrition (*Hay et al., 2014*), and oncology drug development commands billions of dollars in funding each year (*Adams and Brantner, 2006*).

In what follows, we conducted a systematic review and meta-analysis of features of design and outcomes for preclinical efficacy studies of the highly successful drug sunitinib. Sunitinib is a multi-targeted tyrosine kinase inhibitor sunitinib (SU11248, Sutent) and is licensed as monotherapy for three

**eLife digest** Developing a new drug can take years, partly because preclinical research on non-human animals is required before any clinical trials with humans can take place. Nevertheless, only a fraction of cancer drugs that are put into clinical trials after showing promising results in preclinical animal studies end up proving safe and effective in human beings.

Many researchers and commentators have suggested that this high failure rate reflects flaws in the way preclinical studies in cancer are designed and reported. Now, Henderson et al. have looked at all the published animal studies of a cancer drug called sunitinib and asked how well the design of these studies attempted to limit bias and match the clinical scenarios they were intended to represent.

This systematic review and meta-analysis revealed that many common practices, like randomization, were rarely implemented. None of the published studies used 'blinding', whereby information about which animals are receiving the drug and which animals are receiving the control is kept from the experimenter, until after the test; this technique can help prevent any expectations or personal preferences from biasing the results. Furthermore, most tumors were tested in only one model system, namely, mice that had been injected with specific human cancer cells. This makes it difficult to rule out that any anti-cancer activity was in fact unique to that single model.

Henderson et al. went on to find evidence that suggests that the anti-cancer effects of sunitinib might have been overestimated by as much as 45% because those studies that found no or little anti-cancer effect were simply not published. Though it is known that the anti-cancer activity of the drug increases with the dose given in both human beings and animals, an evaluation of the effects of all the published studies combined did not detect such a dose-dependent response.

The poor design and reporting issues identified provide further grounds for concern about the value of many preclinical experiments in cancer. These findings also suggest that there are many opportunities for improving the design and reliability of study reports. Researchers studying certain medical conditions (such as strokes) have already developed, and now routinely implement, a set of standards for the design and reporting of preclinical research. It now appears that the cancer research community should do the same.

different malignancies (*Chow and Eckhardt, 2007*; *Raymond et al., 2011*). As it was introduced into clinical development around 2000 and tested against numerous malignancies, sunitinib provided an opportunity to study a large sample of preclinical studies across a broad range of malignancies—including several supporting successful translation trajectories.

## Results

### Study characteristics

Our screen from database and reference searches captured 74 studies eligible for extraction, corresponding to 332 unique experiments investigating tumor volume response (*Figure 1*, *Table 1*, *Table 1—source data 1E*). Effect sizes (standardized mean difference [SMD] using Hedges' g) could not be computed for 174 experiments (52%) due to inadequate reporting (e.g., sample size not provided, effect size reported as a median, lack of error bars, *Figure 1—figure supplement 1*). Overall, 158 experiments, involving 2716 animals, were eligible for meta-analysis. The overall pooled SMD for all extracted experiments across all malignancies was −1.8 [−2.1, −1.6] (*Figure 2—figure supplement 1*). Mean duration of experiments used in meta-analysis (*Figures 2–4*) was 31 days (±14 days standardized deviation of the mean (SDM)).

### Design elements addressing validity threats

Effects in preclinical studies can fail clinical generalization because of bias or random variation (internal validity), a mismatch between experimental operations and the clinical scenario modeled (construct validity), or idiosyncratic causal mediators in an experimental system (external validity) (*Henderson et al., 2013*). We extracted design elements addressing each using consensus design practices identified in a systematic review of validity threats in preclinical research (*Henderson et al., 2013*).

**A**

| Internal Validity Characteristics | | Experiments (n=158) |
|---|---|---|
| Exact sample size given for all groups | | 120 (76%) |
| Performance of sample size calculation | | 0 (0%) |
| Randomly allocated to treatment | | 58 (37%) |
| Concealed allocation | | 0 (0%) |
| Blinded outcome assessment | | 0 (0%) |
| Used named inferential statistical test | | 117 (74%) |
| Addressed animal flow through experiment | | 21 (13%) |
| Evaluated dose-response (≥3 doses) | | 9 (6%) |
| **Construct Validity Characteristics** | | |
| **Species** | | |
| | Mouse | 156 (99%) |
| | Rat | 2 (1%) |
| **Age** | | |
| | Pediatric/Juvenile | 78 (49%) |
| | Adult | 25 (16%) |
| | Aged | 0 (0%) |
| **Immune Status** | | |
| | Immunocompetent | 30 (19%) |
| | Immunocompromised | 128 (81%) |
| **Sex** | | |
| | Male | 12 (8%) |
| | Female | 104 (66%) |
| **Model Type** | | |
| | Human xenograft | 110 (70%) |
| | Allografts | 42 (27%) |
| | Genetically engineered | 4 (2%) |
| | Other | 2 (1%) |
| **Experiment type** | | |
| | Late-Stage Tumour Model | 63 (40%) |
| | Early-Stage Tumour Model | 91 (58%) |
| **Evidence for causal mechanism** | | |
| | Molecular | 66 (42%) |
| | Physiological | 123 (78%) |
| | Functional/Clinical/Behavioural | 0 (0%) |

**B**

Figure 1. Descriptive analysis of (**A**) internal, construct, and (**B**) external validity design elements. External validity scores were calculated for each malignancy type tested, according to the formula: number species used + number of models used; an extra point was assigned if a malignancy type tested more than one species *and* more than one model.

The following source data and figure supplement are available for figure 1:

**Source data 1**. (A) Coding details for IV and CV categories.

*Figure 1. continued on next page*

*Figure 1. Continued*

**Figure supplement 1**. Descriptive analysis of (**A**) internal, construct, and (**B**) external validity design elements for all experiments (n = 332) extracted for validity data parameters.

Few studies used practices like blinding or randomization to address internal validity threats (*Figure 1A*). Only 6% of experiments investigated a dose–response relationship (3 or more doses). Concealment of allocation or blinded outcome assessment was never reported in studies that advanced to meta-analysis. It is worth noting that one research group employed concealed allocation and blinded assessment for the many experiments it described (*Maris et al., 2008*). However, statistics were reported in a way that did not align with those we needed to calculate SMD. We found that 58.8% of experiments included active drug comparators, thus, facilitating interpretation of sunitinib activity (however, we note that in some of the experiments, sunitinib was an active comparator in a test of a different drug or drug combination). Construct validity practices can only be meaningfully evaluated against a particular, matched clinical trial. Nevertheless, *Figure 1A* shows that experiments predominantly relied on juvenile, female, immunocompromised mouse models, and very few animal efficacy experiments used genetically engineered cancer models (n = 4) or spontaneously arising tumors (n = 0). Malignancies generally scored low (score = 1) for addressing external validity (*Figure 1B*), with breast cancer studies employing the greatest variety of species (n = 2) and models (n = 4).

Implementation of internal validity practices did not show clear relationships with effect sizes (*Figure 3A*). However, sunitinib effect sizes were significantly greater when active drug comparators were present in an experiment compared to when they were not (−2.2 [−2.5, −1.9] vs −1.4 [−1.7, −1.1], p-value <0.001).

Within construct validity, there was a significant difference in pooled effect size between genetically engineered mouse models and human xenograft (p-value <0.0001) and allograft (p-value 0.001) model types (*Figure 3B*). For external validity (*Figure 3C*), malignancies tested in more and diverse experimental systems tended to show less extreme effect sizes (p < 0.001).

**Table 1**. Demographics of included studies

| Study level demographics | Included studies (n = 74) |
| --- | --- |
| Conflict of interest | |
| Declared | 19 (26%) |
| Funding statement* | |
| Private, for-profit | 44 (59%) |
| Private, not-for-profit | 35 (47%) |
| Public | 37 (50%) |
| Other | 2 (3%) |
| Recommended clinical testing | |
| Yes | 37 (50%) |
| Publication date | |
| 2003–2006 | 13 (18%) |
| 2007–2009 | 17 (23%) |
| 2010–2013 | 44 (59%) |

*Does not sum to 100% as many studies declared more than one funding source.

**Source data 1**. (C) Search Strategies. (D) PRISMA Flow Diagram. (E) Demographics of included studies at qualitative level.

## Evidence of publication bias

For the 158 individual experiments, 65.8% showed statistically significant activity at the experiment level ($p < 0.05$, *Figure 2—figure supplement 1*), with an average sample size of 8.03 animals per treatment arm and 8.39 animals per control arm. Funnel plots for all studies (*Figure 4A*), as well as our renal cell carcinoma (RCC) subset (*Figure 4B*) suggest potential publication bias. Trim and fill analysis suggests an overestimation of effect size of 45% (SMD changed from −1.8 [−2.1, −1.7] to −1.3 [−1.5, −1.0]) across all indications. For high-grade glioma and breast cancer, the overestimation was 11% and 52%, respectively. However, trim and fill analysis suggested excellent symmetry for the RCC subgroup, suggesting coverage of the overall effect size and confidence intervals and not overestimation of effect size.

## Preclinical studies and clinical correlates

Every malignancy tested with sunitinib showed statistically significant anti-tumor activity (*Figure 2*). Though we did not perform a systematic review to estimate clinical effect sizes for sunitinib against various malignancies, a perusal of the clinical literature suggests little relationship between pooled effect sizes and demonstrated clinical activity. For instance, sunitinib monotherapy is highly active in RCC patients (*Motzer et al., 2006a*, *2006b*) and yet showed a relatively small preclinical effect; in contrast, sunitinib monotherapy was inactive against small cell lung cancer in a phase 2 trial (*Han et al., 2013*), but showed relatively large preclinical effects.

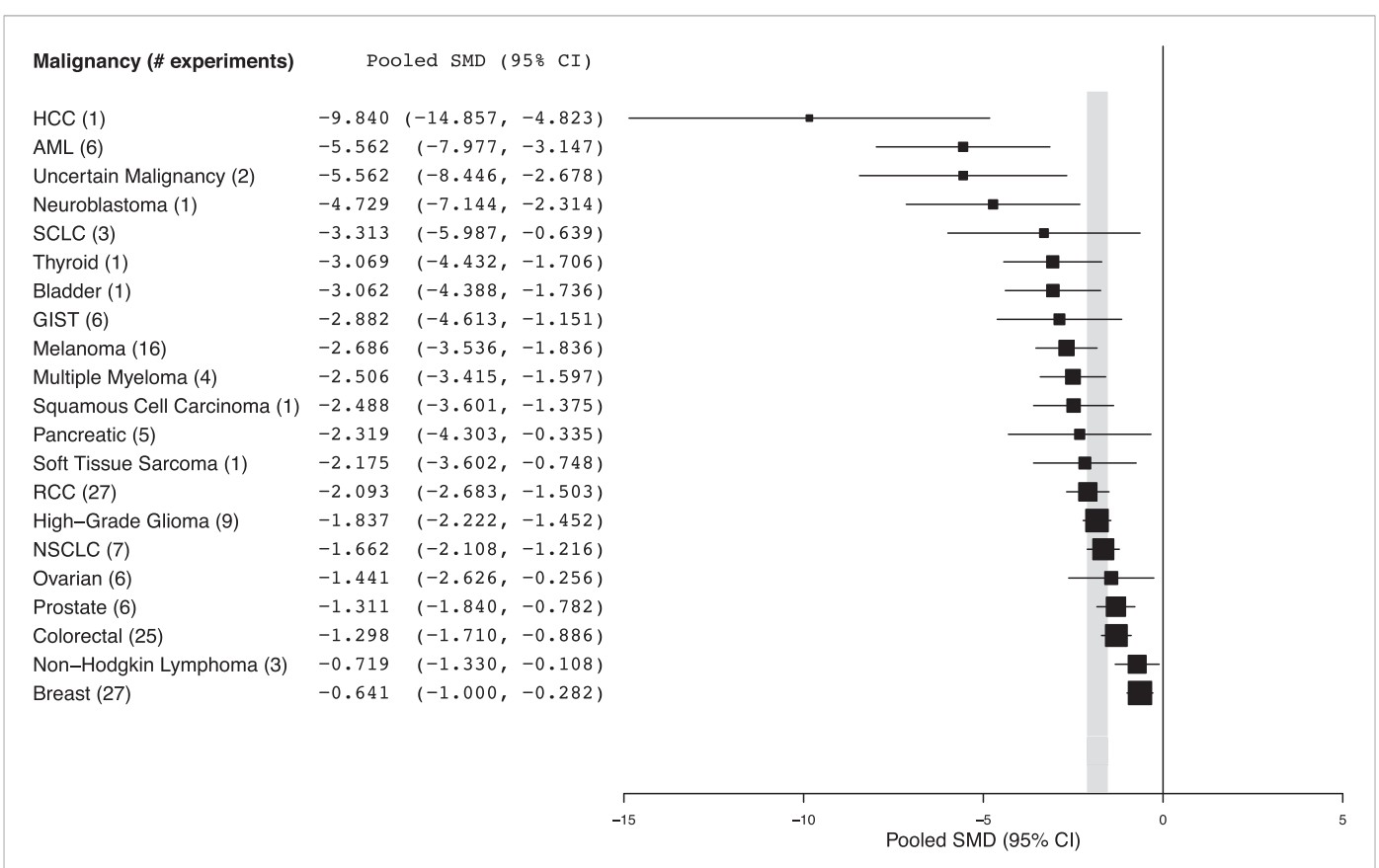

**Figure 2**. Summary of pooled SMDs for each malignancy type. Shaded region denotes the pooled standardized mean difference (SMD) and 95% confidence interval (CI) (−1.8 [−2.1, −1.6]) for all experiments combined at the last common time point (LCT).

The following source data and figure supplement are available for figure 2:

**Source data 1**. (B) Heterogeneity statistics (I²) for each malignancy sub-group.

**Figure supplement 1**. Effect sizes for all included experiments (n = 158).

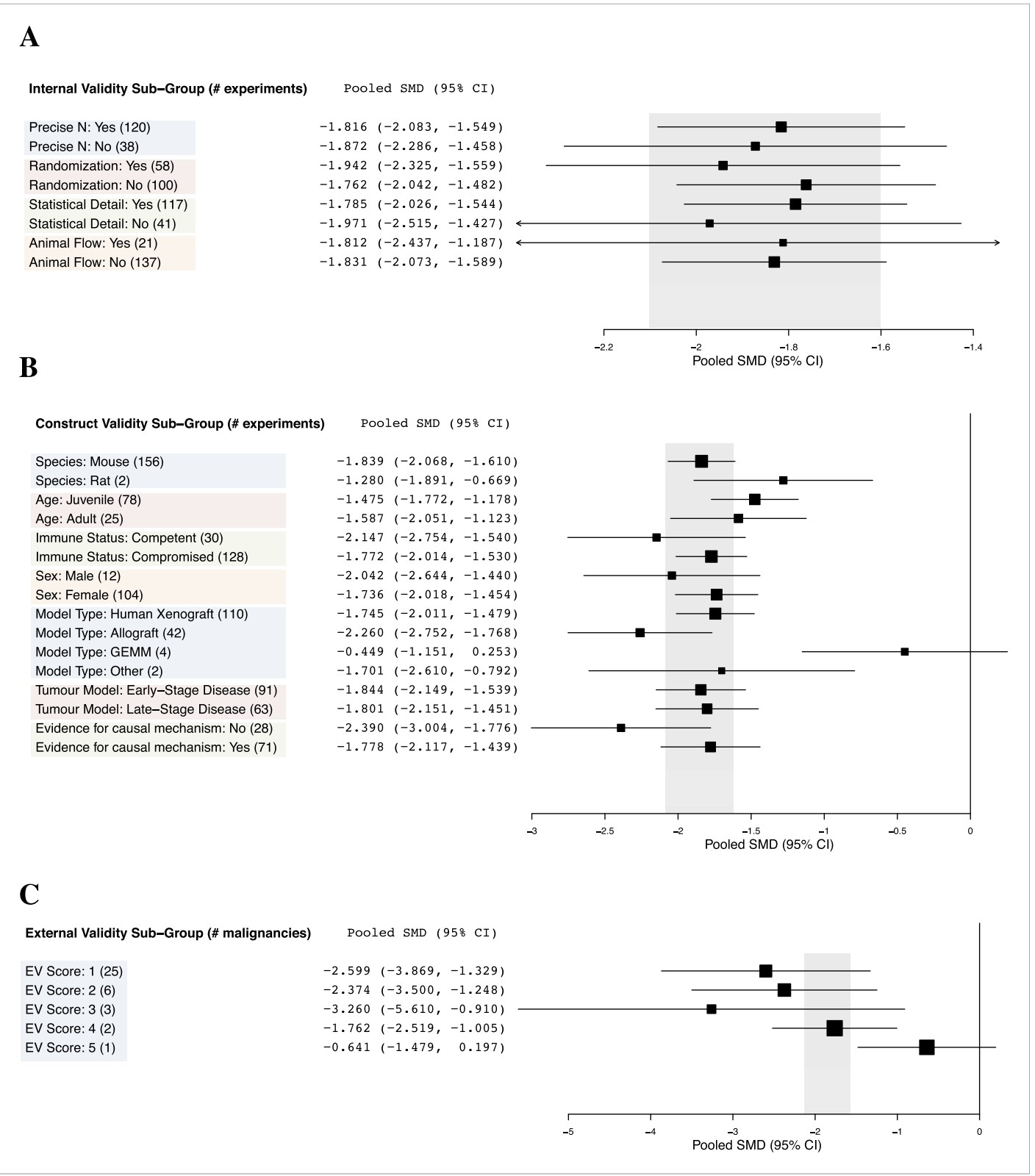

**Figure 3**. Relationship between study design elements and effect sizes. The shaded region denotes the pooled SMD and 95% CI (−1.8 [−2.1, −1.6]) for all experiments combined at the LCT.

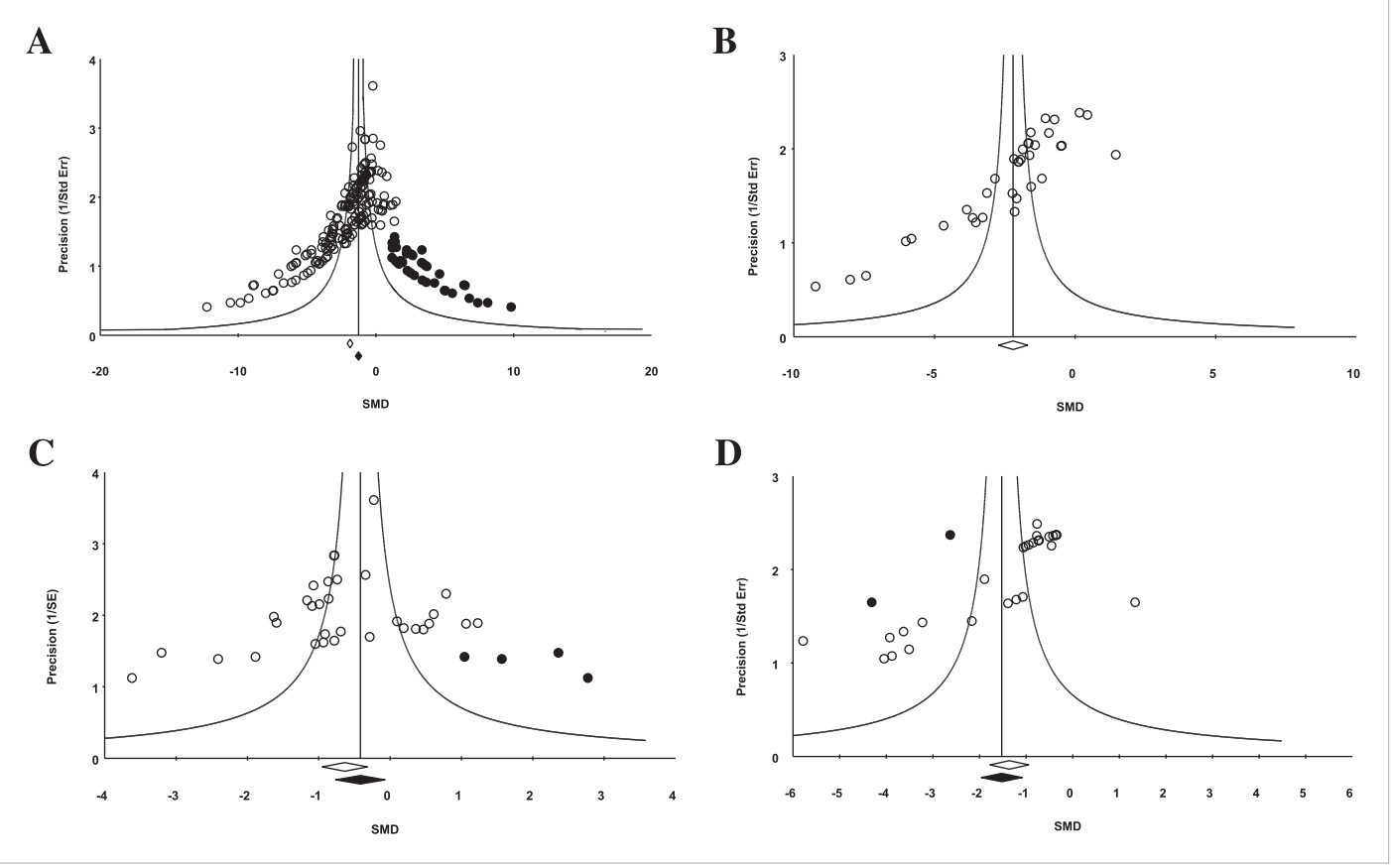

**Figure 4**. Funnel plot to detect publication bias. Trim and fill analysis was performed on pooled malignancies, as well as the three malignancies with the greatest study volume. (**A**) All experiments for all malignancies (n = 182), (**B**) all experiments within renal cell carcinoma (RCC) (n = 35), (**C**) breast cancer (n = 32), and (**D**) colorectal cancer (n = 29). Time point was the LCT. Open circles denote original data points whereas black circles denote 'filled' experiments. Trim and fill did not produce an estimate in RCC; therefore, no overestimation of effect size could be found.

Using measured effect sizes at a standardized time point of 14 days after first administration (a different time point than in *Figures 2–4* to better align our evaluation of dose–response), we were unable to observe a dose–response relationship over three orders of magnitude (0.2–120 mg/kg/day) for all experiments (*Figure 5A*). We were also unable to detect a dose–response relationship over the full dose range (4–80 mg/kg/day) tested in the RCC subset (*Figure 5B*). The same results were observed when we performed the same analyses using the last time point in common between the experimental and control arms.

## Discussion

Preclinical studies serve an important role in formulating clinical hypotheses and justifying the advance of a new drug into clinical testing. Our meta-analysis, which included malignancies that respond to sunitinib in human beings and those that do not, raises several questions about methods and reporting practices in preclinical oncology—at least in the context of one well-established drug.

First, reporting of design elements and data was poor and inconsistent with widely recognized standards for animal studies (*Kilkenny et al., 2010*). Indeed, 98 experiments (30% of qualitative sample) could not be quantitatively analyzed because sample sizes or measures of dispersion were not provided. Experimenters only sporadically addressed major internal validity threats and tended not to test indication-activity in more than one model and species. This finding is consistent with what others have observed in experimental stroke and other research areas (*Macleod et al., 2004*; *van der Worp et al., 2005*; *Kilkenny et al., 2009*; *Glasziou et al., 2014*). Some teams have shown a relationship between failure to address internal validity threats and exaggerated effect size (*Crossley et al., 2008*;

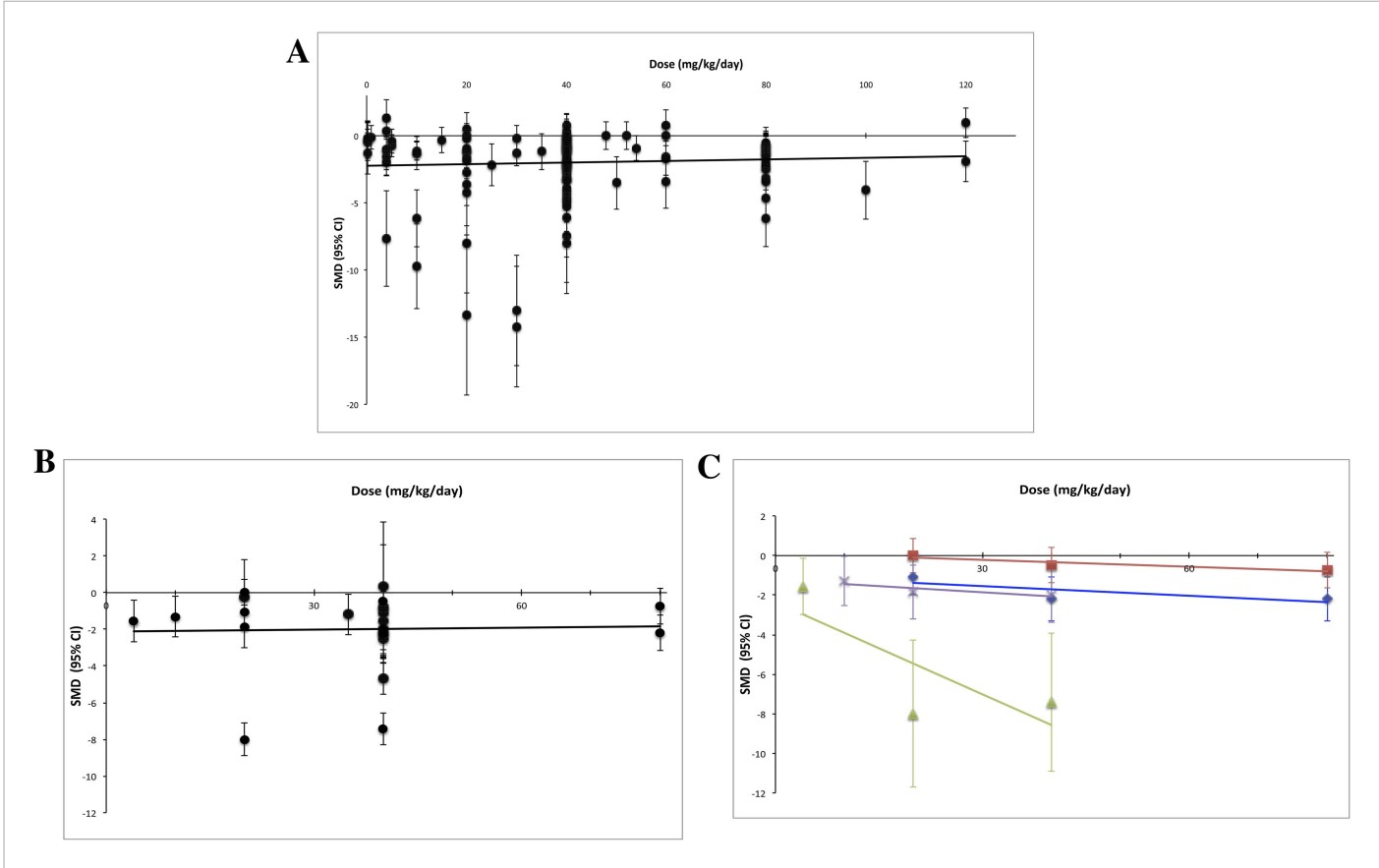

**Figure 5**. Dose–response curves for sunitinib preclinical studies. Only experiments with a once daily (no breaks) administration schedule were included in both graphs. Effect size data were taken from a standardized time point (14 days after first sunitinib administration). (**A**) Experiments (n = 158) from all malignancies tested failed to show a dose–response relationship. (**B**) A dose–response relationship was not detected for RCC (n = 24). (**C**) Dose–response curves reported in individual studies within the RCC subset showed dose–response patterns (blue diamond = Huang 2010a [n = 3], red square = Huang 2010d [n = 3], green triangle = Ko 2010a [n = 3], purple X = Xin 2009 [n = 3]).

*Rooke et al., 2011*); we did not observe a clear relationship. Consistent with what has been reported in stroke (*O'Collins et al., 2006*), our findings suggest that testing in more models tends to produce smaller effect sizes. However, since a larger sample of studies will provide a more precise estimate of effect, we cannot rule out that the trends observed for external validity reflect a regression to the mean.

Second, preclinical studies for sunitinib seem to be prone to publication bias. Notwithstanding limitations on using funnel plots to detect publication bias (*Lau et al., 2006*), our plots were highly asymmetrical. That all malignancy types tested showed statistically significant anti-cancer activity strains credulity. Others have reported that far more animal studies report statistical significance than would be expected (*Wallace et al., 2009*; *Tsilidis et al., 2013*), and our observations that two thirds of individual studies showed significance extends these observations.

Third, we were unable to detect a meaningful relationship between preclinical effect sizes and known clinical behavior. Although a full analysis correlating trial and preclinical effect sizes will be needed, we did not observe obvious relationships between the two. We also did not detect a dose–response effect over three orders of magnitude even within an indication—RCC—known to respond to sunitinib and even when different time points were used. It is possible that heterogeneity in cell lines or strains may have obscured the effects of dose. For example, experimenters may have delivered higher doses to xenografts known to show slow tumor growth. However, RCC patients—each of whom harbors genetically distinct tumors—show dose–response effects in trials (*Faivre et al., 2006*) and between trials in a meta-analysis (*Houk et al., 2010*). It is also possible that the toxicity of sunitinib may have limited the ability to demonstrate dose response, though this

contradicts demonstration of dose response within studies (*Abrams et al., 2003*; *Amino et al., 2006*; *Ko et al., 2010*). Finally, the tendency for preclinical efficacy studies to report drug dose, but rarely drug exposure (i.e., serum measurement of active drug), further limits the construct validity of these studies (*Peterson and Houghton, 2004*).

One explanation for our findings is that human xenograft models, which dominated our meta-analytic sample, have little predictive value, at least in the context of receptor tyrosine kinase inhibitors. This is a possibility that contradicts other reports (*Kerbel, 2003*; *Voskoglou-Nomikos et al., 2003*). We disfavor this explanation in light of the suggestion of publication bias; also, xenografts should show a dose–response regardless of whether they are useful clinical models. A second explanation is that experimental methods are so varied as to mask real effects. However, we note that the observed patterns on experimental design are based purely on what was reported in 'Materials and methods' section. Third, experiments assessing changes in tumor volume might only be interpretable in the context of other experiments within a preclinical report, such as with mechanistic and pharmacokinetic studies. This explanation is consistent with our observation that studies testing effect along a causal pathway tended to produce smaller effect sizes. A fourth possible explanation for our findings is that the predictive value of a small number of preclinical studies was obscured by inclusion of poorly designed and executed preclinical studies in our meta-analysis. Quantitative analysis of preclinical design factors that confer greater clinical generalizability awaits side-by-side comparison with pooled effects in clinical trials. Finally, it may be that design and reporting practices are so poor in preclinical cancer research as to make interpretation of tumor volume curves useless. Or, non-reporting may be so rampant as to render meta-analysis of preclinical research impossible. If so, this raises very troubling questions for the publication economy of cancer biology: even well-designed and reported studies may be difficult to interpret if their results cannot be compared to and synthesized with other studies.

Our systematic review has several limitations. First, we relied on what authors reported in the published study. It is possible certain experimental practices, like randomization, were used but not reported in methods. Further to this, we relied only on published reports, and restriction of searches to the English language may have excluded some articles. In February of 2012, we filed a Freedom of Information Act request from the Food and Drug Administration (FDA) for additional preclinical data submitted in support of sunitinib's licensure; nearly 4 years later, the request has not been honored. Second, effect sizes were calculated using graph digitizer software from tumor volume curves: minor distortion of effect sizes may have occurred but were likely non-differential between groups. Third, subtle experimental design features—not apparent in 'Materials and methods' sections—may explain our failure to detect a dose–response effect. For instance, few reports provide detailed animal housing and testing conditions, perhaps leading to important inter-laboratory differences in tumor growth. It should also be emphasized that our study was exploratory in nature; findings like ours will need to be confirmed using prespecified protocols. Fourth, our study represents analysis of a single drug, and it may be our findings do not extend beyond receptor tyrosine kinase inhibitors, or sunitinib. However, many of our findings are consistent with those observed in other systematic reviews of preclinical cancer interventions (*Amarasingh et al., 2009*; *Sugar et al., 2012*; *Hirst et al., 2013*). Fifth, our analysis does not directly address many design elements—like duration of experiment or choice of tissue xenograft—that are likely to bear on study validity. Finally, we acknowledge that there may be funding constraints that limit implementation of validity practices described above. We note, nevertheless, that other realms, in particular, neurology, have found ways to make such methods a mainstay.

Numerous commentators have raised concerns about the design and reporting of preclinical cancer research (*Sugar et al., 2012*; *Begley, 2013b*). In one report, only 11% preclinical cancer studies submitted to a major biotechnology company withstood in-house replication (*Begley and Ellis, 2012*). The Center for Open Science and Science Exchange has initiated a project that will attempt to reproduce 50 of the highest impact papers in cancer biology published between 2010 and 2012 (*Morrison, 2014*). In a recent commentary, Smith et al. fault many researchers for performing in vitro preclinical tests using drug levels that are clinically unachievable due to toxicity (*Smith and Houghton, 2013*). Unaddressed preclinical validity threats like this—and the ones documented in our study—encourage futile clinical development trajectories. Many research areas, like stroke, epilepsy, and cardiology, have devised design guidelines aimed at improving the clinical generalizability of preclinical studies (*Fisher et al., 2009*; *Galanopoulou et al., 2012*; *Curtis et al., 2013*; *Pusztai et al., 2013*); and the ARRIVE guidelines (*Kilkenny et al., 2010*) for reporting animal experiments have been

taken up by numerous journals and funding bodies. Our findings provide further impetus for developing and implementing guidelines for the design, reporting, and synthesis of preclinical studies in cancer.

# Materials and methods

## Literature search

To identify all in vivo animal studies testing the anti-cancer properties of sunitinib ('efficacy studies'), we queried the following databases on 27 February 2012 using a search strategy adapted from *Hooijmans et al. (2010)* and *de Vries et al. (2011)*: Ovid MEDLINE In-Process & Other Non-Indexed Citations and Ovid MEDLINE (dates of coverage from 1948 to 2012), EMBASE Classic and EMBASE database (dates of coverage from 1974 to 2012) and BIOSIS Previews (dates of coverage from 1969 to 2012). Search results were entered into an EndNote library and duplicates were removed. Additional citations were identified during the screening of identified articles. See *Table 1—source data 1C,D* for detailed search strategy and PRISMA flow diagram.

Screening was performed at citation level by two reviewers (CF and VCH), and at full-text by one reviewer (VCH). Inclusion criteria were (a) original reports or abstracts, (b) English language, (c) contained at least one experiment measuring disease response in a live, non-human animals, and (d) employed sunitinib in a control, comparator, or experimental context, (e) tested anti-cancer activity. To avoid capturing the same experiment twice, in rare cases where the same experiment was reported in different articles, the most detailed and/or recent publication was included.

## Extraction

All included studies were evaluated at the study-level, but only those with eligible experiments (e.g., those evaluating the effect of monotherapy on tumor volume and that were reported with sample sizes and error measurements) were forwarded to experiment-level extractions. We excluded experiments when they had been reported in a previous publication after specifically searching for duplicates during screening and analysis. For each eligible experiment, we extracted experimental design elements derived from a prior systematic review of validity threats in preclinical research (*Henderson et al., 2013*).

Details regarding the coding of internal and construct validity categories are given in *Figure 1—source data 1A*. To score for external validity, we created an index that summed the number of species and models tested for a given malignancy and awarded an extra point if more than one species *and* model was tested. For example, if experiments within a malignancy tested two species and three different model types, the external validity score would be 4 (1 point for the second species, one point for the second model type, one point for the third model type, and an extra point because more than one model and species were employed).

Our primary outcome was experimental tumor volume and we extracted necessary information (sample size, mean measure of treatment effect, and SDM/SEM) to enable calculation of study and aggregate level effect sizes. Since the units of tumor volume were not always consistent between experiments, we extracted those experiments for which a reasonable proxy of tumor volume could be obtained. These included physical caliper measurements (often reported in $mm^3$ or $cm^3$), tumor weights (often reported in mg), optical measurements made from luminescent tumor cell lines (often reported in photons/second), and fold differences in tumor volumes between the control and treatment arms. We extracted experiments of both primary and metastatic tumors, but not experiments where tumor incidence was reported. To account for these different measures of tumor volume, SMDs were calculated using Hedges' g. Hedges' g is a widely accepted standardized measure of effect in meta-analyses where units are not always identical. For experiments where more than one dose of sunitinib was tested against the same control arm, we created a pooled SMD to adjust appropriately for the multiple use of the same control group. Data were extracted at baseline (Day 0 and defined as the first day of drug administration), Day 14 (the closest measured data point to 14 days following first dose), and the last common time point (LCT) between the control group and the treatment group. The LCT was variable between experiments and the last time point for which we could calculate SMD and often represented the point at which the greatest difference was observed between the arms. Data presented graphically were extracted using the graph digitizer software GraphClick (Arizona Software). Extraction was performed by four independent and trained coders

(VCH, ND, AH, and NM) using DistillerSR. There was a 12% double-coding overlap to minimize inter-rater heterogeneity and prevent coder drift. Discrepancies in double coding were reconciled through discussion, and if necessary, by a third coder. The gross agreement rate before reconciliation for all double-coded studies was 83%.

## Meta-analysis

Effect sizes were calculated as SMDs using Hedges' g with 95% confidence intervals. Pooled effect sizes were calculated using a random effects model employing the *DerSimonian and Laird (1986)* method, in OpenMeta[Analyst] (*Wallace et al., 2009*). We also calculated heterogeneity within each malignancy using $I^2$ statistics (*Figure 2—source data 1B*). To assess the predictive value of preclinical studies in our sample, we calculated pooled effect sizes for each type of malignancy. Subgroup analyses were performed for each validity element. p-values were calculated by a two-sided independent group T-test. Statistical significance was set at a p-value <0.05; as this was an exploratory study we did not adjust for multiple analyses.

Funnel plots to assess publication bias and Duval and Tweedie's trim and fill estimates were generated using Comprehensive Meta Analyst software (*Dietz et al., 2014*). Funnel plots were created for all experiments in aggregate, and for the three indications for which greater than 20 experiments were analyzable.

Dose–response curves are a widely used tool for testing the strength of causal relationships (*Hill, 1965*), and if preclinical studies indicate real drug-responses, we should be able to detect a dose–response effect across different experiments. Dose–response relationships were found in post-analysis of sunitinib clinical studies in metastatic RCC and Gastrointestinal stromal tumour (GIST) (*Houk et al., 2010*). We tested for all indications in aggregate, as well as for RCC, an indication known to respond to sunitinib in human beings (*Motzer et al., 2006a*, *2006b*, *2009*). To eliminate variation at the LCT between treatment and control arms, dose–response curves were created using data from a time point 14 days from the initiation of sunitinib treatment. Experiments with more than one treatment arm were not pooled as in other analyses, but expanded out so that each treatment arm (with it's respective dose) could be plotted properly. As we were unable to find experiments that reported drug exposure (e.g., drug serum levels), we calculated pooled effect sizes in OpenMeta[Analyst] and plotted against dose. To avoid the confounding effect of discontinuous dosing, we included only experiments that used a regular administration schedule without breaks (i.e., sunitinib administered at a defined dose once a day instead of experiments where sunitinib was dosed more irregularly or only once).

As this meta-analysis was exploratory and involved development of methodology, we did not prospectively register a protocol.

## Acknowledgements

Dan G Hackam, Jeremy Grimshaw, Malcolm Smith, Elham Sabri, Benjamin Carlisle.

## Additional information

### Funding

| Funder | Grant reference | Author |
|---|---|---|
| Canadian Institutes of Health Research (Instituts de recherche en santé du Canada) | EOG 111391 | Jonathan Kimmelman |

The funder had no role in study design, data collection and interpretation, or the decision to submit the work for publication.

### Author contributions

VCH, JK, Conception and design, Acquisition of data, Analysis and interpretation of data, Drafting or revising the article; ND, AH, NMK, CAF, Acquisition of data, Drafting or revising the article; DF, Conception and design, Analysis and interpretation of data, Drafting or revising the article

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
