## [Decision Letter]

Thank you for submitting your work entitled “A Meta-Analysis of Validity Threats and Clinical Correlates in Preclinical Research of Sunitinib” for peer review at *eLife*. Your submission has been favorably evaluated by Prabhat Jha (Senior Editor) and three reviewers, one of whom, Dawn Teare, is a member of our Board of Reviewing Editors. The other two reviewers, Carlijn Hooijmans and Glenn Begley, have agreed to share their identity.

The reviewers have discussed the reviews with one another and the Reviewing Editor has drafted this decision to help you prepare a revised submission.

The authors describe a meta-analysis of preclinical studies performed with sunitinib. Their analysis found poor scientific methodology with for example lack of blinding, lack of evidence of power calculations and likely bias in over-reporting of positive results. It provides objective evidence to support claims that have been made by others. This is an important and valuable contribution. It addresses an issue of increasing scientific and societal importance.

Essential revisions:

1) The dose response analysis needs revision. The systematic review has collected the data from animal research but I cannot see where the clinical outcome data comes from. In fact I do not see any clinical data. Is the assumption that the clinical data does show a dose response? I find the lack of a dose response rather curious in that presumably individual studies did report an effect but somehow there is heterogeneity between the studies and hence this reduces to a random scatter? There appear to be a lot of points in Figure 5 but I cannot easily see how many studies were eligible for the dose-response. I also do not understand the statistical analysis going along with Figure 5. What sort of curve/line is being fitted? What role does the two-group *t*-test have in a dose response analysis? The line drawn on the figure appears to be a straight line. I think this section needs to be more clearly written.

2) It would be very helpful for the authors to comment on the number of studies that reported drug exposure rather than drug dose. It is common for preclinical studies to report the intended drug dose, but without reference to the actual levels of drug attained. Assessment of drug levels confirms (i) that drug was actually administered and (ii) to the intended cohort. Moreover, it is frequently the case that when levels rather than dose are examined, it is evident that the levels achieved in rodents are not tolerated in humans. This has specifically been recognized for sunitinib and is an additional explanation for the disconnection between preclinical versus clinical effectiveness.

3) It would also be helpful if the authors could comment on the number of studies that included a positive control. While sunitinib may have had activity, without a positive control it is difficult to place that activity into context. Was the efficacy comparable or greater than that seen with another known active agent?

4) It would be beneficial to understand how many of the studies used an appropriate statistical test when analyzing their data.

5) The authors did not comment on the ARRIVE Guidelines for pharmacology studies. These should be discussed (cited in Begley and Ioannidis., Reproducibility in Science. Improving the Standard for Basic and Preclinical Research., Circulation Research., 2015; 116: 116-126.)

6) The authors should check that their references are correctly cited. For example, the statement (in the Introduction) “A recent systematic review of guidelines for limiting bias in preclinical research design was unable to identify any guidelines in oncology (20)” is not supported by [20].

7) The observation that “Our findings suggest that testing in more models tends to produce smaller effect sizes” has also been made previously (cited in [6]).

8) Why did the authors of this review decide a) to include only papers published in English b) not to retrieve data via contacting the authors of the original papers? The authors conclude that there is a substantial risk on publication bias. But it might be possible that the observed risk on publication bias could have been much smaller when the authors contacted the authors for missing data and included also papers published in languages other than English.

9) In this review, the outcome measure of interest needs a more detailed description in the Material and methods section. In the subsection “Extraction,” the authors state that they are only interested in tumor volume (“[…] those evaluating the effect of monotherapy on tumor volume”), whereas the Abstract and Introduction talk about tumor growth (which is a much broader term). Did the authors only assess tumor volume of primary tumors? Or did they include the volumes of metastases in the body as well? Did the authors include studies in which the weight of tumors and metastases were assessed? And what about tumor incidence? The authors also state: “[…] To account for different measures of tumor growth”; which measures of tumor growth did they include?

10) More details are needed in the Materials and methods concerning the meta analyses:

a) Are data pooled in case tumor volume was determined in various regions of the body within one experiment?

Are the data pooled in the overall analyses (Figure 3) when results were assessed at various time points?

b) In animal studies the same control group is often used for multiple experimental. Did the authors correct for multiple use of the control groups?

c) What was the minimum group size to allow subgroup analyses?

11) The authors should report group sizes of all subgroups (also in figures; e.g. Figure 3), and take group size into account when interpreting the results.

12) The authors should report heterogeneity statistics (for example I^2^), and take these results into account when interpreting the data.

---

## [Author Response]

*Essential revisions*:

*1) The dose response analysis needs revision. The systematic review has collected the data from animal research but I cannot see where the clinical outcome data comes from. In fact I do not see any clinical data. Is the assumption that the clinical data does show a dose response? I find the lack of a dose response rather curious in that presumably individual studies did report an effect but somehow there is heterogeneity between the studies and hence this reduces to a random scatter? There appear to be a lot of points in*
Figure 5
*but I cannot easily see how many studies were eligible for the dose-response. I also do not understand the statistical analysis going along with*
Figure 5*. What sort of curve/line is being fitted? What role does the two-group* t*-test have in a dose response analysis? The line drawn on the figure appears to be a straight line. I think this section needs to be more clearly written*.

Dose response is discussed twice in the manuscript. First it appears when we report the fraction of studies that tested a dose response. Here, our emphasis is on dose response testing as a means of enhancing the internal validity of claims regarding clinical promise. The second mention is at the end of our Results section. There, the dose-response analysis was not meant to compare our preclinical results directly with sunitinib clinical data (although a dose-response relationship in GIST and mRCC has been demonstrated by others – see below). Our premise was that preclinical studies should demonstrate a dose-response relationship in the absence of overwhelming inter-experimental heterogeneity and publication bias. This premise was based on the known dose response in human beings, and the fact that dose-response was shown within individual preclinical experiments. The reviewers have interpreted our point correctly: our inability to show a clear dose-response after pooling all results raises some concerns about the predictive value of studies included in our meta-analysis.

Regarding sample size, there are 158 data points in Figure 5. Each data point represents a single experiment or sub-experiment (in cases where one experiment had more than one treatment arm with a defined dose). Inclusion into the dose-response analysis required experiments or sub-experiments to have used a regular dosing schedule (i.e. sunitinib administered at a defined dose once a day instead of experiments where sunitinib was dosed more irregularly or only once). These inclusion criteria led to 26 experiments or sub-experiments to be excluded. It is worth reiterating that the number of data points possible in our dose-response analysis for all malignancies was 183. This number differs from the total number of data points possible in our other analyses, where we calculated a weighted average of the SMDs and 95% CIs in the cases of experiments that had more than one treatment arm.

Regarding the curve we fitted, we believe our simple regression (calculated using the linear regression function in Excel) is sound for descriptive purposes and assessing a simple relationship between doses. Our intent was simply to judge the presence or absence of an expected positive correlation between effect and increasing dose rather than identifying precise dose-response curves.

Regarding the source of clinical outcome data, we think this confusion stems from a poor choice of title, which we changed to “A Meta-Analysis of Threats to Valid Clinical Inference in Preclinical Research of Sunitinib”.

We modified the manuscript as follows:

We enhanced the description of dose response in Methods, Results, and Discussion sections. We also cite Houk et al (PMID: 19967539) to show that a dose-response relationship was found in GIST and mRCC in clinical studies. This citation helps support our assertion that the absence of a dose-response in preclinical models (especially when RCC was analyzed alone) is odd.

We also removed the two-group *t*-test, which was a remnant of a previous draft and out of place.

We made some edits to figure legend, as well as Methods section, to explain more clearly how (and why) we performed dose response curves, as well as the number of experiments included in them.

*2) It would be very helpful for the authors to comment on the number of studies that reported drug exposure rather than drug dose. It is common for preclinical studies to report the intended drug dose, but without reference to the actual levels of drug attained. Assessment of drug levels confirms (i) that drug was actually administered and (ii) to the intended cohort. Moreover, it is frequently the case that when levels rather than dose are examined, it is evident that the levels achieved in rodents are not tolerated in humans. This has specifically been recognized for sunitinib and is an additional explanation for the disconnection between preclinical versus clinical effectiveness*.

We agree that reporting drug exposure (and measured serum levels of drug in experimental animals) would be the optimal measurement upon which to construct dose-response curves and to later correlate with clinical data. Unfortunately, these measurements were rarely included within tumour volume experiments we extracted. Indeed, this comment prompted us to reexamine the first 20% of studies in our sample and we found no instances where levels were measured instead of dose. We had included 4 studies that included serum measurements of sunitinib post-dosing (PK). However, none of these serum levels were taken during efficacy experiments; they were only performed in studies to correlate with molecular data (e.g. FLT3 expression levels).

We protocolized and standardized eligibility criteria and dose exposure measurement to reduce variability inherent with using dose, given that drug serum levels were not available to us. For instance, in dose response analysis, we only included experiments where the dose was clearly stated and was administered once per day (as opposed to more complex or variable dosing schedules). Additionally, we used a common time point between experiments (14 days after the first day of drug administration).

The difficulty we faced constructing a robust measure of drug exposure and effect further highlights the poor reporting practices in preclinical research and the need for improvement in this regard.

In light of the above, we made the following modifications to the manuscript:

We added a statement in the Methods section as to why we used dose instead of drug exposure (“As we were unable to find experiments that reported drug exposure (e.g. drug serum levels), we calculated pooled effect sizes in OpenMeta[Analyst] and plotted against dose”).

We also added a statement in the Discussion section stating “the tendency for preclinical efficacy studies to report drug dose, but rarely drug exposure (i.e. serum measurement of active drug), further limits the construct validity of these studies,” and we buttressed this with a new reference (Peterson et al., 2004).

3) It would also be helpful if the authors could comment on the number of studies that included a positive control. While sunitinib may have had activity, without a positive control it is difficult to place that activity into context. Was the efficacy comparable or greater than that seen with another known active agent?

We agree that efficacy data are difficult to make sense of absent positive controls. Examining our dataset, we found that 93/158 experiments included in the meta-analysis included an active comparator drug arm. Note that we do not use the term “positive control” strictly here, as sometimes sunitinib was being the positive control in a study of a newer drug.

We inserted two sentences to the subsection “Design Elements Addressing Validity Threats” describing the frequency of active drug comparators in experiments, and the relationship between their inclusion and reported effect sizes.

*4) It would be beneficial to understand how many of the studies used an appropriate statistical test when analyzing their data*.

We agree this would be a useful analysis but it is beyond the scope of this paper. To address appropriateness adequately we would need raw subject data, subject flow, protocols, and analytical plans. While some issues are black and white, many are nuanced/subjective and require complete background info. Instead, we focused our energies in the validity criteria identified in our PLoS Medicine systematic review of preclinical design guidelines.

*5) The authors did not comment on the ARRIVE Guidelines for pharmacology studies. These should be discussed (cited in Begley and Ioannidis., Reproducibility in Science. Improving the Standard for Basic and Preclinical Research., Circulation Research., 2015; 116: 116-126*.*)*

We heartily agree that citing ARRIVE is important for this audience and we’ve now cited and singled it out for mention at the end of the Discussion. We also added a reference to the Circulation Research article, which provides a nice overview of reproducibility challenges. We also took this occasion to mention the Center for Open Science Reproducibility Project: Cancer Biology.

*6) The authors should check that their references are correctly cited. For example, the statement (in the Introduction) “A recent systematic review of guidelines for limiting bias in preclinical research design was unable to identify any guidelines in oncology (*[20]*)” is not supported by*
[20].

Thanks to the reviewers for catching this. It has now been corrected.

*7) The observation that “Our findings suggest that testing in more models tends to produce smaller effect sizes” has also been made previously (cited in*
[6]*)*.

Thank you. We now cite the Circulation article (see above). For this particular claim, the referees are correct – this has been suggested before – in particular by CAMARADES. We now cite the paper (O’Collins et al, Annals Neurol, 2006).

*8) Why did the authors of this review decide a) to include only papers published in English b) not to retrieve data via contacting the authors of the original papers? The authors conclude that there is a substantial risk on publication bias. But it might be possible that the observed risk on publication bias could have been much smaller when the authors contacted the authors for missing data and included also papers published in languages other than English*.

This was decided mainly on feasibility and funding constraints. We did not have a budget to both identify relevant expertise and to translate articles. Contacting authors for additional data would pull in experiments and findings that have not been subject to peer review. Our experience contacting drug companies for missing data does not lead us to think this would have been a productive use of our time. We did, however, attempt to obtain Sugen/Pfizer’s preclinical dataset from FDA by filing a Freedom of Information Act. The FOIA was submitted in February of 2012, around the time the grant received funding. The 3-year grant ended in 2015, with the FOIA request not yet honoured. We added a sentence in the Discussion stating this as a limitation.

*9) In this review, the outcome measure of interest needs a more detailed description in the Material and methods section. In the subsection “Extraction,” the authors state that they are only interested in tumor volume (“[…] those evaluating the effect of monotherapy on tumor volume”), whereas the Abstract and Introduction talk about tumor growth (which is a much broader term). Did the authors only assess tumor volume of primary tumors? Or did they include the volumes of metastases in the body as well? Did the authors include studies in which the weight of tumors and metastases were assessed? And what about tumor incidence? The authors also state*: *“[…] To account for different measures of tumor growth”; which measures of tumor growth did they include?*

We expanded the Methods section to reflect the reviewers’ queries. For instance, we changed our broad references to “tumour growth” into references to tumour volume measurements (which is what we were extracting). We also provide a more detailed description of outcome measurement (see, for example, text beginning “Our primary outcome was experimental […]”, in the subsection “Extraction”).

*10) More details are needed in the Materials and methods concerning the meta analyses*.

a) Are data pooled in case tumor volume was determined in various regions of the body within one experiment?

We were a bit unsure what the reviewers mean by “various regions of the body within one experiment”. We did not encounter studies where tumors were injected into more than one location within a single experiment. Perhaps the referees are referring to inter-experimental variability in tumour implantation site (e.g. experimental renal tumour injected subcutaneously into the hindleg vs. injected orthotopically into the renal capsule of the mouse). If this is the case, we did record this information (as to whether the tumour implantation site was heterotopic, orthotopic, or systemic), but we did not perform separate analyses according to this variable. Our concern is that doing so would divide our sample into slices too thin to enable analysis. If referees think it useful we would be happy to add a sentence or two about the use orthotopic vs. heterotopic models in our sample of experiments.

*Are the data pooled in the overall analyses (*Figure 3*) when results were assessed at various time points?*

We clarified information of our time points to the Methods (in the subsection “Extraction”). We also clarified which time point was used for which analysis in the Results section and in each figure legend. As explained in the subsection “Study Characteristics”, the mean number of days in last common timepoint (LCT) was 31 days (± 14 days SDM).

b) In animal studies the same control group is often used for multiple experimental. Did the authors correct for multiple use of the control groups?

Indeed, we did adjust. In cases where multiple arms of treatment were reported versus the same control, a pooled estimate of the treatment arms was calculated using the inverse variance approach and then compared to the single control to produce an SMD estimate and 95% CIs.

c) What was the minimum group size to allow subgroup analyses?

We did not specify a minimum group size for meta-analysis. Instead we presented all available data. The goal of subgroup analyses in meta-analysis is to present all available data relevant to the strata even if just 1 experiment, thus a minimum subgroup size is not as relevant. Now that we have integrated sample sizes into our figures more clearly, readers can perhaps make their own judgments.

*11) The authors should report group sizes of all subgroups (also in figures; e.g.*
Figure 3*), and take group size into account when interpreting the results*.

We converted Figure 3 to forest plots and have included the subgroup sizes.

*12) The authors should report heterogeneity statistics (for example I*^*2*^*), and take these results into account when interpreting the data*.

The primary purpose of our paper was to evaluate potential sources of methodological and pre-clinical heterogeneity through pre-specified items of internal, external, and construct validity. We now report I^2^ statistics but we must note that we (and others – e.g. Paul SR, Donner A., Small sample performance of tests of homogeneity of odds ratios in k 2×2 tables., Stat Med 1992; 11: 159-65; Hardy RJ, Thompson SG., Detecting and describing heterogeneity in meta-analysis., Stat Med 1998; 17: 841-56) are somewhat skeptical about the value of heterogeneity statistics due to their extreme sensitivity in the presence of a small number of studies. As the referees will see, heterogeneity statistics, as expected, within each malignancy was generally high (see [Supplementary-material SD3-data]), with some notable exceptions (e.g. high-grade glioma, prostate cancer). By design, our study sheds light on possible sources of methodological heterogeneity (i.e. Figure 3) and we have incorporated such observations in our interpretation and conclusions.